# Recreational Drug Misuse and Its Potential Contribution to Male Fertility Levels’ Decline: A Narrative Review

**DOI:** 10.3390/brainsci12111582

**Published:** 2022-11-19

**Authors:** Nicolò Schifano, Stefania Chiappini, Alessio Mosca, Andrea Miuli, Maria Chiara Santovito, Mauro Pettorruso, Paolo Capogrosso, Federico Dehò, Giovanni Martinotti, Fabrizio Schifano

**Affiliations:** 1Department of Urology, ASST Sette Laghi–Circolo e Fondazione Macchi Hospital, 21100 Varese, Italy; 2Psychopharmacology, Drug Misuse and Novel Psychoactive Substances Research Unit, School of Life and Medical Sciences, University of Hertfordshire, Hertfordshire AL10 9EU, UK; 3Department of Neuroscience, Imaging and Clinical Sciences, “G. D’Annunzio” University, 66100 Chieti, Italy

**Keywords:** substance abuse, drug abuse, substance dependence, male infertility

## Abstract

Recreational drug intake may be associated with a range of medical untoward consequences, including male infertility. However, as the related evidence is still limited, the main outcome of this review is to provide a better understanding of the existence of any association between recreational drug misuse and male fertility levels’ decline. Whilst searching the MEDLINE/PubMed, a comprehensive overview of the literature regarding male infertility and substances of abuse (e.g., phytocannabinoids, opiates/opioids, stimulants, ‘herbal highs’, psychedelics, and ‘novel psychoactive substances) was here undertaken. Due to the paucity of robust, high-quality, empirical, human studies, a narrative strategy was here preferred over a systematic approach. Relevant data are qualitatively analyzed and presented in a table. Although most available evidence is in support of a detrimental role of cannabis on human spermatogenesis, a few remaining studies failed to document any effect of this drug on seminal quality, and it is not clear to which extent this drug impacts fertility rates/time to pregnancy. The current understanding of the impact of opiate-, cocaine- and amphetamine/stimulant-misuse on human reproduction is widely unknown, and most studies dealing with this matter represent only an extrapolation of data derived from specific clinical circumstances. Although the message of ‘no smoking, no alcohol and no street drugs’ should always be offered as good health advice to all patients seeking medical help for fertility issues, robust scientific clinical evidence in support of a direct detrimental impact of recreational drugs on spermatogenesis is scant to date.

## 1. Introduction

### 1.1. Psychoactive Substances’ Intake and Related Medical Untoward Consequences; Male Infertility: Definition, Causes, and Risk Factors

Recreational drug intake may be associated with a range of medical untoward consequences. For example, a few cardiotoxic (e.g., dysrhythmias, cardiac arrest, chest pain, and myocardial infarction) effects have been reported for those molecules interacting with the cannabinoid receptor e.g., phytocannabinoids [1], and especially so for the high-potency synthetic cannabimimetics [2,3]. Conversely, the use of ‘uppers’ (including cocaine, synthetic cathinones, and amphetamine-type stimulants) is typically associated with both agitation, ranging from mild agitation to severe psychosis, and a vast range of sympathomimetic effects (e.g., tachycardia and hypertension as well as psychoactive effects): hyperthermia, rhabdomyolysis, renal failure, and seizures [2]. Finally, users of the 5-HT2A agonist psychedelics report effects such as euphoria, mild stimulation, enhanced appreciation of music/light, visually appealing distortions, and altered sense of time and space [2,4]. 

One of the least-mentioned untoward consequences of recreational drug misuse is, however, its possible association with male infertility. Infertility can be defined as the failure to achieve spontaneous pregnancy after one year of unprotected intercourse [5]. Recent evidence seems to suggest that there may be an overall decline in male fertility levels, and especially so in the Western world, where about 15% of the couples are infertile and seek medical treatment for infertility [5]. Considering that up to one-half of infertile couples may have a male factor present [6], investigators continue to focus on identifying the etiology of male fertility decline. Environmental factors and lifestyle have been associated with alterations in sperm production and may potentially have an impact on male fertility. A number of studies have linked tobacco smoking habit and alcohol consumption with seminal quality impairment, with this association being even stronger for heavy smoking and heavy drinking, and especially so when these habits are concomitant [7]. Similarly, in a large cohort [8] of more than 1200 young healthy men, habitual moderate alcohol intake was associated with a reduction in semen quality, although the decreasing trend was most significant for men with a typical weekly intake above 25 units. In addition, recent alcohol intake (e.g., during the week prior to the visit) was associated with an increase in serum testosterone, potentially secondary to alcohol-induced changes in the liver metabolism of testosterone itself. Among the other potentially reversible factors which may have a detrimental impact on male fertility, the lack of physical activity, increased body mass index (BMI) and adiposity, and unhealthy diets have been associated with poor semen quality [9,10,11,12]. Reducing the exposure to lifestyle risk factors is currently considered the main intervention for improving sperm quality in men with sub-fertility, and clinicians should always collect a history of adverse lifestyle factors in all couples presenting with levels of sub-fertility [5].

### 1.2. Drug Use as a Risk Factor for Male Infertility

One of the additional etiological factors which has been hypothesized to have a potential impact on male fertility may well be constituted by recreational drug intake. Drug abuse, as well as smoking and alcohol consumption, is frequently seen among patients seeking medical attention for uro-andrological issues, as some of these recreational compounds are well-known for their impact on male erectile function and ejaculation [13]. In a recent large-sized retrospective series [13], data from 7447 men presenting to an uro-andrological clinic were analyzed. The reasons for their presentation to medical attention were lower urinary tract symptoms (LUTS), erectile dysfunction (ED), and male factor infertility in 25.7%, 39.5%, and 34.8% of the cases, respectively. Previous drug misuse was reported by 378 (5.1%) men, and 190 (2.6%) individuals were current users. Interestingly enough, those patients seeking uro-andrological advice for infertility were more frequently presenting with drug-misusing issues (107; 4.1%) vs. those patients presenting with ED (66; 2.2%) or LUTS (17; 0.9%) (both *p* < 0.001). Moreover, current drug misusers more frequently reported nicotine smoking (*p* < 0.001) and alcohol consumption (*p* < 0.001) than those without a history of drug misuse.

Recreational drug misuse is a widespread phenomenon; the COVID-19 pandemic has facilitated its increasing levels in the general population, and a diversification of the products has been made available, especially from the web [14]. According to the most recent European Monitoring Centre for Drug and Drug Addiction (EMCDDA) report [15], cannabinoids, stimulants such as cocaine and methamphetamines, opiates/opioids, and psychedelics still represent the most popular drugs being self-administered with a recreational purpose by a range of subjects, and especially so by youngsters, e.g., those most likely contributing to the fertility levels of the general population. The use of these illicit drugs is common in Europe, with a yearly prevalence rate for any drug consistently higher in males compared with females [15].

In view of possible ethical issues, any interventional study dealing with the effects of recreational drug misuse is typically not feasible in the clinical context, and therefore most human studies focusing on these issues are retrospective. Unfortunately, this methodological approach introduces a number of potential confounders and biasing factors, which are difficult to control. Conversely, preclinical studies present with the advantage of allowing a better evaluation of the effect of the drug exposure on the animal model, but these studies should be interpreted with caution due to a range of reasons, including higher drug-exposure levels in these models vs. human subjects.

The current narrative review aims at describing and critically appraising the most recent suggestions of the literature relating to the potential effect on male fertility of the most popular recreational molecules, e.g., cannabinoids, stimulants, opiates/opioids, psychedelics, ‘herbal highs’, and novel psychoactive substances (NPS).

Regarding alcohol, the molecule may indeed be a component of the recreational drug scene, where it is being ingested in a poly drug consumption pattern. Conversely, alcohol intake is largely prevalent in the general, non-drug-misusing population. Furthermore, a range of high-quality empirical studies focusing on the impact of low/moderate/high alcohol intake levels [16,17] have already dealt with this issue. For these reasons, alcohol was not here considered as part of the focus of the current review.

## 2. Materials and Methods

### 2.1. Screening and Selection Process

A comprehensive search was performed on MEDLINE/PubMed, since its inception, to identify those studies which are here considered as both appropriate and representative. The search itself, which was carried out between May and October 2022, focused on those studies using the terms ‘male infertility’, ‘recreational drug misuse’, ‘cannabinoids’, phytocannabinoids’, ‘opiates’, ‘opioids’, ‘amphetamines’, ‘amphetamine-type stimulants’, ‘stimulants’, ‘cocaine’, ‘methylphenidate’, ‘stimulant plants and herbs’, ‘herbal highs’, ‘LSD’, ‘psychedelics’, ‘novel psychoactive substances’, ‘NPS’, AND ‘male infertility’. A range of keyword strings were used, e.g., drugs AND male infertility, infertility OR among healthy individuals AND recreational drug misuse/abuse/dependence. Resulting abstracts were scrutinized for relevance; the related studies were then summed up after an interactive peer-review process. More in particular, the related papers were here selected for their appropriateness by NS and SC; any disagreement was resolved with the help of a senior researcher (FS). 

### 2.2. Inclusion and Exclusion Criteria

Only articles published in English in peer-reviewed journals were selected. Inclusion criteria related to both quantitative and qualitative studies relating to the use of recreational drugs in humans. Although publications from the past 10 years (e.g., from October 2012 to October 2022) were given the highest levels of priority in the selection process, a number of highly regarded older publications were here included as well. Conversely, studies focusing on children or subjects with medical diagnoses using the selected drugs/substances for medical reasons were excluded from the review.

### 2.3. Data Analysis and Presentation

With the current strategy, some 98 papers were here identified; however, for 49 of them, the main focus referred to molecules (e.g., nicotine, anabolic androgenic steroids, prescribed medications, and alcohol) different from those identified here as being of interest. Conversely, the remaining 49 (e.g., opiates/opioids: 13; cocaine: 1; amphetamine-type substances/remaining stimulants: 4; THC: 27; herbal highs: 2; LSD: 2; and NPS: 0) related to the main focus of the review. Interestingly, some 43/49 of these studies were published over the last decade. However, due to the paucity of robust, high-quality, empirical human studies, a narrative strategy was here preferred over a systematic approach. Relevant data were here qualitatively analyzed, and most representative findings are presented in Table 1. 

## 3. Results

### 3.1. Cannabis

Marijuana use is increasingly common, and concerns of further legalization in a number of countries across the globe prompt the collection of more precise information regarding its potential impact on the reproductive function itself. The possibly deleterious effects of cannabis on male fertility remain under-investigated, with cannabis recreational users being typically unaware of its possible implications on health, including the spermatogenetic function. A recently published cross-sectional survey in a Canadian population of infertile men [18] found that 13% of the men had used cannabis in the past year; 38% had previously used cannabis more than 1 year before the survey; and only 49% had never used cannabis. Last year, users perceived less potential negative effects on reproductive health compared to previous (e.g., >1 year users) and never users (*p* values < 0.05). 

Δ9—tetrahydrocannabinol—is the main active compound of cannabis, which binds to the related cannabinoid receptors (Figure 1). 

These receptors are represented both in the brain and in the testes; therefore, cannabis may exert both direct (e.g., on the testes) and indirect (e.g., on the hypothalamic–pituitary–gonadal axis) effects on spermatogenesis. The presence of cannabinoid receptors, such as CB1, in human testicles, vas deferens, and sperm cells [19] suggests a likely regulatory role of the endocannabinoids in the physiology of the spermatogenetic process. An indirect effect of cannabis on spermatogenesis has been documented in animal models through down-regulation of the hypothalamic–pituitary–gonadal axis with consequent androgen suppression and hyperprolactinaemia [20], but also direct spermatotoxicity secondary to oxidative stress [21,22] and consequent poor sperm quality. Testicular histological changes were also demonstrated in terms of reduction of Johnsen score in rats [20]. A direct toxic effect on spermatogenesis through the oxidative mechanism has been documented as well in a recent retrospective cross-sectional study [23], where 27 infertile regular cannabis users were matched with 27 infertile cannabis non-users. The authors investigated the potential of cannabis to elicit sperm nuclear alterations using a range of different investigations, including fluorescence in situ hybridization (FISH) to assess numerical chromosomal abnormalities and terminal deoxynucleotidyl-transferase-mediated (dUTP) nick-end labeling to investigate DNA fragmentation. The rates of sperm aneuploidy, diploidy, chromosome abnormalities and DNA fragmentation were significantly (all at *p* < 0.05) higher in cannabis users vs. controls.

The clinical evidence in human studies of the detrimental effects of cannabis on spermatogenesis are mainly based on seminal and hormonal parameters’ data. The vast majority of these studies recruited patients with a history of either drug abuse or infertility, thus introducing possible concerns regarding the generalizability of the findings. Hehemann et al. [24] prospectively evaluated the semen analyses from 409 men presenting to medical attention for infertility; 17% of them were current cannabis users presenting with an increased risk of abnormal sperm morphology [odds ratio (OR) 2.15, 95% confidence interval (CI): 1.21–3.79] and decreased semen volume according to the WHO reference standards (OR 2.76, 95%CI: 1.19–6.42). Similarly, Carroll et al. [25] investigated the impact of cannabis use on the semen quality of a population of Jamaican patients attending the infertility clinic for assisted reproduction purpose. About one-half of the participants disclosed cannabis use, with recent- and large-volume users being respectively 2.6 and 4.3 times at greater odds of being diagnosed with asthenozoospermia (all *p* < 0.05). Moderate-quantity users were 3.4 times more likely to be diagnosed with an abnormal morphology (teratozoospermia). Gundersen et al. [26] instead were the first to examine the effect of cannabis use on semen quality from healthy young men in the general population; 45% of the total number of those 1215 men who were recruited disclosed recent cannabis use (e.g., in the past 3 months), and 25% of them reported a frequent (e.g., more than once per week) intake. Sperm concentration, total sperm count, percentage of motile sperm, and percentage of morphologically normal forms were all decreased among the frequent users. Together with cannabis, 10.9% of the recruited subjects had used different recreational drugs (e.g., amphetamine, ecstasy, cocaine) as well, showing even greater declining levels in terms of seminal quality parameters. Significant odds of worse seminal parameters were confirmed, even after the stratified analyses (e.g., these were performed to identify subjects at highest risk and to mitigate the possible confounders including co-existent unhealthy lifestyles, such as tobacco smoking, alcohol usage, elevated BMI) were carried out. No differences in luteinizing hormone (LH) or follicle-stimulating hormone (FSH) were found, but frequent marijuana users had 7% higher testosterone levels, which was at odds with the previous findings. Notwithstanding the documented impact of cannabis on seminal quality, the authors failed to show a statistically significant difference in terms of the odds of presenting with altered seminal parameters according with the fourth and fifth editions of the World Health Organization (WHO) reference manuals on semen analyses [27]. Conversely, Nassan et al. [28] in a cross-sectional cohort study, including 365 cannabis users vs. 297 controls, showed higher sperm concentration levels in cannabis users. Men who had smoked marijuana had a significantly higher sperm concentration (62.7 (95% confidence interval: 56.0, 70.3) million/mL) than men who had never smoked marijuana (N = 297) (45.4 (38.6, 53.3) million/mL) after adjusting for potential confounders (*p* = 0.0003). There were no significant differences in sperm concentration between current and past marijuana smokers. In these results, being at odds with previous findings, the authors cautioned that their data would not be generalizable to men from the general population. Finally, Belladelli et al. [29] recently investigated the evidence on this matter with the help of a meta-analysis of those studies, investigating the possible differences in both seminal parameters and hormonal profiles in men exposed to cannabis vs. non-users. They failed to demonstrate any statistically significant effects of cannabis use in terms of both semen quality and reproductive hormones’ levels.

### 3.2. Opiates/Opioids

The rise in the misuse of opiates/opioids is becoming a health challenge globally. Opiate/opioids’ misuse is an overlooked and concerning phenomenon which most typically has an iatrogenic origin, as it begins with unnecessary/inappropriate drug prescription [30]. Indeed, these molecules may impair male spermatogenesis via different pathways; androgen suppression with associated hypogonadism may be associated with a direct gonadotropin release inhibition and/or with a hypothalamic–pituitary–gonadal axis impairment secondary to opioid-induced prolactin release. Opiates, in fact, suppress the hypothalamic dopamine secretion, thus reducing the inhibitory impact of the dopamine on the release of prolactin from the anterior pituitary gland (Figure 2). 

A number of human studies, but not all [31], have reported increased levels of prolactin in opioid abusers. Opioids may also exert a direct effect on testosterone levels through their interaction with the opioid receptors in the testes, where they can influence the expression of genes involved in the antioxidant defense system [32]. Reduced epithelial thickness in seminiferous tubules was observed in a number of preclinical studies in rats treated with opiates [32,33,34,35]. Additionally, initial in vitro studies [36] showed that the kappa–opioid receptor (KOR) is involved in the regulation of the fertilizing capacity of human sperm. The KOR activation causes the translocation of a protein called SPANX-A/D into the nucleus of the sperm cell, where this protein probably regulates sperm production.

Human studies documented that opioid treatment resulted in decreased testosterone levels, whilst no significant impact was observed on prolactin and gonadotropins [37,38], with the level of androgens returning back to normal when the opioids’ administration was reduced [39]. In a recent case control study [40], twenty-four healthy men were matched with the same number of heroin-addicted men; seminal pH, white blood cell count in semen and sperm motility were significantly different between the groups. Additionally, the seminal miRNA-122 levels in addicted men (3.51 ± 0.73) were statistically higher than in healthy men (1.52 ± 0.54) (*p* = 0.034). The authors concluded that heroin abuse may be associated with male infertility due to leukocytospermia, asthenozoospermia, protamine deficiency, and seminal plasma miRNA profile alteration. Nazmara et al. [41] collected semen and blood samples from 25 heroin-addicted men and the same number of healthy men, identifying lower sperm motility and viability levels in heroin-addicted men. Additionally, sperm histone replacement abnormalities were more frequent in the addicted group, whilst androgen levels were similar between groups. Safarinejad et al. [42] evaluated the impact of opioid usage on semen quality and sperm DNA integrity in 142 opiate addicted vs. 146 healthy controls; they identified significantly lower sperm concentration rates and significant increase in the amount of fragmented DNA in opiate users vs. controls. Not only heroin, but also other opiates which are typically used for their analgesic properties, can exert an unfavorable effect on spermatogenesis when misused. In a recent case-control study [43], 60 patients with a history of tramadol abuse were matched with 30 healthy controls. Tramadol misusers had higher prolactin and lower testosterone levels and were significantly more likely to have lower sperm count and higher abnormal motility of sperm compared with controls.

### 3.3. Cocaine

Cocaine is a commonly used drug, with a male predominance in its use. It functions by increasing levels of the neurotransmitter dopamine in brain circuits related to the control of movement and reward. Normally, dopamine recycles in the cell that released it, disrupting the signal between nerve cells. However, cocaine prevents dopamine from being recycled, causing large amounts to accumulate in the space between two nerve cells, disrupting their normal communication, which clearly strongly reinforcing drug-taking behavior (Figure 3).

The very limited knowledge regarding the effects of cocaine on spermatogenesis derives from animal studies, and the mechanism whereby this molecule may exert deleterious effects on male fertility remains largely unknown, even though it appears to be mediated by sperm cell apoptosis [44,45,46]. George et al. [47] documented that rats receiving high dosages of cocaine presented with a decrease in seminiferous tubule diameter and a significantly lower pregnancy rate of 33% vs. 86%. Similarly, another group [48] confirmed the previous findings of morphometric testicular structure changes in rats treated with cocaine, with a disruption of spermatogenesis and a significant decrease in the diameter and number of the seminiferous tubules; decreased cell adhesions; abnormal cellular structures; and overall decreased quantity of germ cells. Evidence from human studies of the cocaine potential impact on male fertility is very limited; indeed, the relatively low drug use rates in infertile populations vs. other more commonly used drugs, and the subjects’ reporting bias, make it difficult to investigate the effects of cocaine use on male fertility. Although one would conclude that a clear understanding of the true, direct, effect of cocaine on human males’ fertility remains unknown, the classical observations of Bracken et al. [49] suggest decreased sperm concentration, motility, viability and morphology in chronic cocaine abusers. Samplaski et al. [50] instead aimed to evaluate the impact of cocaine use on a population of infertile men presenting for fertility evaluation. Some 38 patients (0.9%) out of the total number of recruited male subjects reported recent cocaine use. These subjects, were however, significantly more likely to have been exposed to additional different recreational drugs (e.g., marijuana, heroin, ecstasy, lysergic acid diethylamide (LSD), and anabolic steroids for bodybuilding); tobacco; and sexually transmitted infections (STIs), such as chlamydia and/or herpes.

### 3.4. Amphetamine-Type Stimulants (ATS)

ATS are a large group of substances, including amphetamine, methamphetamine, and MDMA/ecstasy-like compounds, which are used recreationally and can be taken orally, injected or smoked. Both methamphetamine and MDMA/ecstasy are two of the five most reported ‘sexualized’ drugs, e.g., somehow involved/facilitating the sexual encounter, increasing the release of dopamine (Figure 4) [51]. 

Amphetamines have been reported to decrease sperm count, normal morphology and motility [52], and circulatory testosterone levels [53], and induce apoptosis in seminiferous tubules in mice [54]. The detrimental effects of methamphetamines on spermatogenesis may be the result of their interaction with the gamma-aminobutyric acid (GABA)-ergic pathway [55], secondary to the compensatory upregulation of GABA production and its functions in testes. Methamphetamine was also proved to impair spermatogenesis through an interference with the glycolysis, since the expression of a number of key-metabolites of this pathway (e.g., glucose transporter 1 (GLUT1)) was found to be impaired in the exposed mice [55]. Aryan et al. [56] examined the expressions of a number of markers of inflammation from the testes of 10 methamphetamine users obtained during autopsies and compared these data with those identified from testes obtained from autopsies of a reference group of 10 healthy men. Testicular samples from methamphetamine users showed higher levels of reactive oxygen species (ROS) and a decrease in anti-oxidant activity. 

Methylphenidate is the most widely prescribed molecule for attention-deficit hyperactivity disorder (ADHD). The prescription rates of methylphenidate increased in the United Kingdom by almost 800% between the years 2000 and 2015, and the treatment rate in males was five times higher than in females [57]. Although a number of preclinical studies [58,59] examined the impact of methylphenidate on spermatogenesis, to date, only a few studies have investigated the effect of this medication on human semen quality parameters. Pham et al. [60] performed a retrospective cohort study where patients were stratified by exposure to stimulants (e.g., either methylphenidate or amphetamines). Stimulant use was associated with decreased total motile sperm count and decreased semen volume, whilst both sperm concentration and morphology were found to be non-significantly different in exposed vs. non-exposed men. Finally, Shalev et al. [61] found that methylphenidate exposure did not affect sperm morphology and was paradoxically associated with increased sperm concentration rates in current and previous methylphenidate users vs. methylphenidate-naïve patients.

### 3.5. Novel Psychoactive Substances (NPS); ‘Herbal Highs’; and Psychedelics

Although the diffusion of novel psychoactive substances (NPS) and herbal psychoactive substances is a widespread and increasing phenomenon [62], the evidence of their possible impact on spermatogenesis is extremely limited and mostly confined to animal studies [63], or case reports [64].

Hakim et al. [65] assessed a total number of 214 male patients with an history of infertility and khat abuse (i.e., alone or in combination with tobacco, coffee and alcohol intake), identifying no statistically significant impact of this exposure on seminal quality vs. non-drug users.

No other robust human studies were identified in the literature specifically dealing with the possible impact of novel psychoactive substances (NPS), herbal highs and psychedelics, such as LSD, on male fertility.

## 4. Discussion

Despite the range of intriguing data presented, this paper may have provided here only a preliminary assessment of data on the association between drug misuse and male infertility. Overall, it seemed from here that misusing with recreational compounds, and especially so with cannabinoids [25], may putatively lead to the impairment of hypothalamic-pituitary–gonadal functions, increased sperm DNA fragmentation and apoptosis, and reduced sperm quality [66]. 

The current understanding of the impact of opiate/opioid-, cocaine- and ATS misuse on human reproduction is largely unknown, and the vast majority of the studies dealing with this matter may just represent an extrapolation of information derived from specific clinical circumstances and are not directly drawn from either the general population or the population of infertile patients. Overall, although evidence from human studies on this matter is extremely limited, one could still tentatively hypothesize a potential negative impact of these molecules on semen quality [67].

### 4.1. Male Infertility Measurement; Methodological Issues

Most empirical studies are mainly focused here on semen analysis (e.g., sperm concentration, motility, viability, and morphology), but this biological parameter, per se, is not an ideal discriminator of male fertility. Furthermore, as was very recently suggested, there is a lack of knowledge about possible molecules used as candidates for the early identification of male infertility risk [68]. It is of interest to note that the different studies here commented did refer to a range of different parameters to define male infertility (e.g., abnormal morphology, abnormal motility, low sperm volume, presence of asthenozoospermia and/or teratozoospermia, white blood cell count in semen, survival rate of sperm cells, etc.; see Table 1), and this may have introduced here some biases and related difficulties in the overall interpretation of findings. 

In their milestone study on the impact of cannabis use on male fertility, Gundersen et al. [26], in contrast with Carroll et al. [25], failed to show a statistically significant difference in the probability of having altered seminal parameters, as being defined by the WHO reference standards. Given these findings, one could argue that there are levels of conflicting evidence relating to the effects of cannabis abuse on spermatogenesis. It is however necessary to keep in mind that drug abusers are different from non-users from a range of perspectives, including higher odds of unhealthy lifestyle habits such as alcohol, tobacco use, and caffeine [28]. For example, the relationship between caffeine consumption and infertility in the general population is unclear; hence, Bu et al. [69] carried out a systematic review of the evidence from any type of controlled clinical studies to explore whether caffeine intake is a risk factor for human infertility. Their results showed that low (OR 0.95, 95%CI 0.78–1.16), medium (OR 1.14, 95%CI 0.69–1.86) and high doses (OR 1.86, 95%CI 0.28–12.22) of caffeine intake may not increase the risk of infertility. They suggested as well, however, that the quality of the current evidence bodies was low. In contrast with this, a systematic review assessing the role of caffeine intake on male infertility concluded that caffeine intake, possibly through sperm DNA damage, may negatively affect male reproductive function [70]. The authors suggested as well, however, that the evidence from epidemiological studies on semen parameters and fertility is inconsistent and inconclusive. Although regression sub-analyses can be used to estimate the impact of these confounders, residual biasing influences cannot be completely controlled for. Whether or not these semen quality changes may actually have an impact on fertility rates still remains to be conclusively determined. A retrospective analysis [71] of the cross-sectional survey data from the National Survey of Family Growth aimed to determine if cannabis use may have an impact on time to pregnancy. No statistically significant impact of marijuana use on time to pregnancy was documented among a total number of 758 male and 1076 female responders, with 16.5% of males and 11.5% of females having reported cannabis use while attempting to conceive.

### 4.2. Strengths and Limitations

Overall, satisfactory numbers of homogeneous human-subject-related papers were not here identified. Due to this paucity of data, in order to fulfil the purpose of this review, a narrative approach was here chosen. Indeed, with respect to a systematic review approach, this narrative review may have provided broader literature coverage and more flexibility. We acknowledge, however, that this design may have introduced a range of limitations. Explicit criteria for article selection have not been disclosed for some of the studies here included, leading to possible selection biases. The review strategy adopted resulted in the inclusion of studies which were highly heterogeneous in their design, methodologic quality, and patient’s features. Furthermore, the data may be influenced by publication bias, as studies that report negative or null associations often go unpublished. Finally, we acknowledge that this review did not cover the full range of substances of abuse (including alcohol, nicotine, and prescribing medications) which may be putatively responsible for deleterious effects on spermatogenesis. To mitigate these issues, a structured approach was used to identify the most recent relevant studies on this topic, and a comprehensive range of original articles was here selected and included.

## 5. Conclusions

Both drug misuse and male infertility are on the increase globally, and concerns of a possible drug-induced male sub-fertility are to be taken into serious account. Although the message of ‘no smoking, no alcohol and no street drugs’ should always be offered as good health advice to all patients seeking medical help for fertility issues [72], robust scientific clinical evidence in support of a direct detrimental impact of recreational drugs on spermatogenesis is scant to date. Despite drug misuse possibly significantly contributing to alter spermatogenesis and seminal quality, delaying assisted conception to make poorly evidenced changes to lifestyle is unlikely to enhance conception and may indeed be prejudicial in couples with little time to lose.

Finally, in agreement with Alghobary and Mostafa [73], it is here felt that future studies should include the use of logistic regression analysis to better understand the effect of specific substances on human male fertility.

## Figures and Tables

**Figure 1 brainsci-12-01582-f001:**
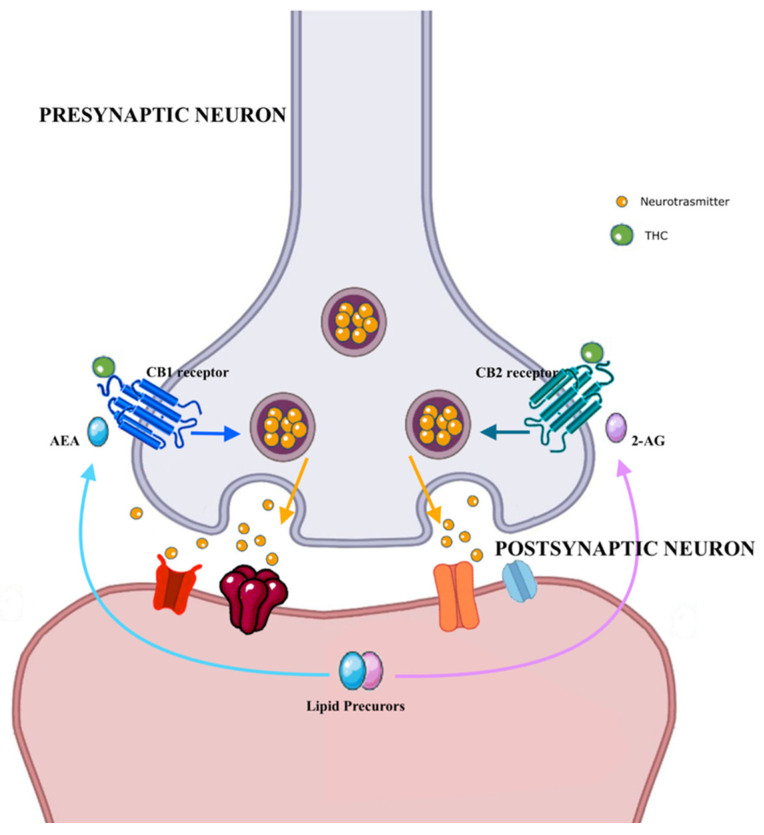
Molecular mechanisms associated with cannabis effects. Abbreviations: AEA: anandamide, CB1 receptor: cannabinoid 1 receptor, CB2 receptor: cannabinoid 1 receptor, THC: tetrahydrocannabinol.

**Figure 2 brainsci-12-01582-f002:**
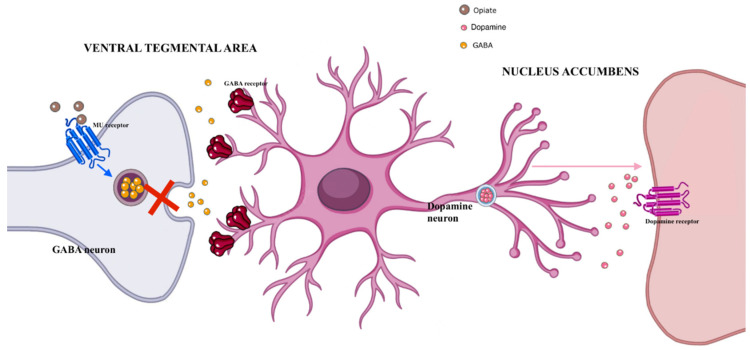
Molecular mechanisms associated with opioid effects. Abbreviations: GABA: gaba aminobutyric acid.

**Figure 3 brainsci-12-01582-f003:**
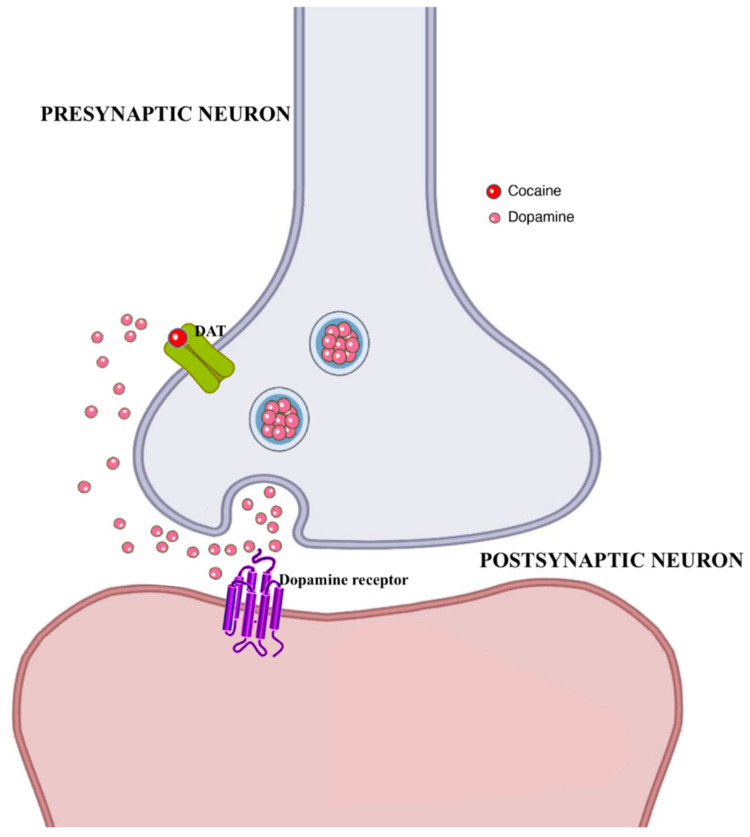
Molecular mechanisms associated with cocaine effects. Abbreviations: DAT: dopamine transporter.

**Figure 4 brainsci-12-01582-f004:**
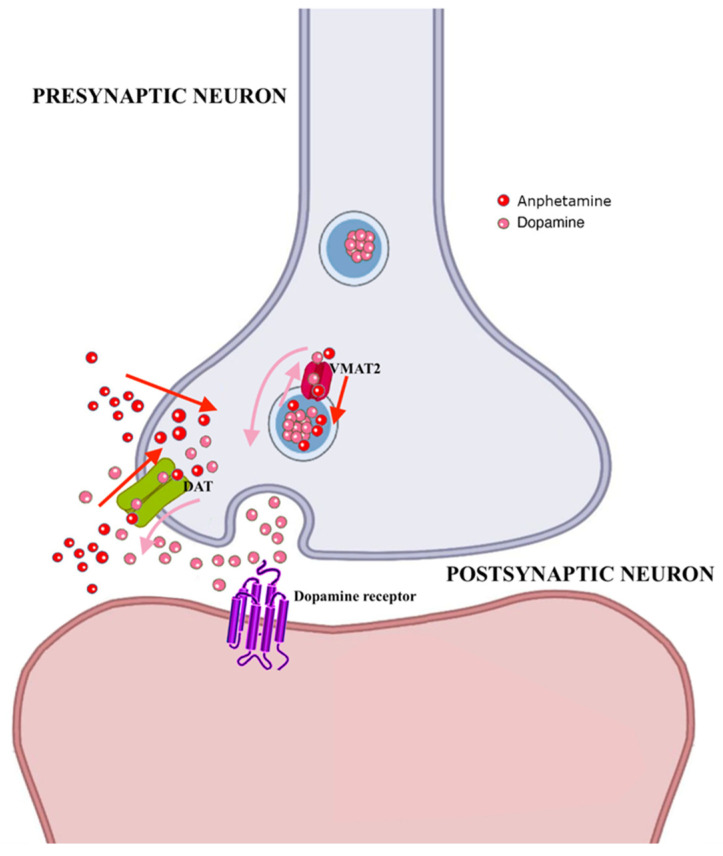
Molecular mechanisms associated with amphetamine effects. Abbreviations: DAT: dopamine transporter, VMAT2: vesicular monoamine transporter 2.

**Table 1 brainsci-12-01582-t001:** Recreational drug (e.g., cannabis; opiates/opioids; amphetamine-type stimulants; and khat) misuse and its impact on male fertility levels: summary of most representative findings.

Drug	Author	Year	Country	Study Design	Sample Size	Population Description	Outcomes Being Analyzed	Main Findings
Cannabis	Hehemann et al.	2021	U.S.	Prospective	409	71 current users 103 past users	Semen quality	Current use was associated with increased odds of abnormal morphology [OR 2.15, 95%CI: 1.21–3.79] and abnormal semen volume (OR 2.76, 95%CI: 1.19–6.42), while odds of less than WHO reference value sperm motility were reduced (OR 0.47, 95%CI: 0.25–0.91)
Cannabis	Carroll et al.	2020	Jamaica	Cross-sectional	229	47% current users	Semen quality	Recent- and large-quantity use were 2.6 times (OR 2.6; 95%CI, 1.0–6.8) and 4.3 times (OR 4.3; 95%CI, 1.1–15.9) at greater risk of asthenozoospermia. Moderate quantity users were 3.4 times (OR 3.4; 95%CI, 1.5–7.9) more likely to be diagnosed with teratozoospermia
Cannabis	Gundersen et al.	2015	Denmark	Prospective	1215	45% recent users	Semen quality, reproductive hormone levels	Regular marijuana smoking associated with a 28% (95%CI: −48, −1) lower sperm concentration and a 29% (95%CI: −46, −1) lower total sperm count. The combined use of marijuana and other recreational drugs reduced the sperm concentration by 52% (95%CI: −68, −27) and total sperm count by 55% (95%CI: −71, −31). Marijuana smokers had higher levels of testosterone
Cannabis	Nassan et al.	2019	U.S.	Cross-sectional	662	365 patients never used cannabis	Semen quality, reproductive hormone levels	Men who had ever smoked marijuana had significantly higher sperm concentration (62.7 million/mL (95%CI: 56.0, 70.3)) than men who had never smoked marijuana ((45.4 million/mL (38.6, 53.3)). No significant differences in sperm concentration between current and past marijuana smokers. A similar pattern was observed for total sperm count. Marijuana smokers had significantly lower FSH concentrations than never marijuana smokers
Opiates/opioids	Nazmara et al.	2020	Iran	Case-control	48	24 addicted 24 healthy controls	Semen quality, levels of protamine-2gene and miRNA-122 in seminal plasma	White blood cell count in semen (1.69 ± 0.41 vs. 8.61 ± 1.73, *p* = 0.001), motility (65.51 ± 2.57 vs. 41.96 ± 3.58, *p* = 0.001) and survival rate (87.41 ± 1.00 vs. 71.50 ± 4.59, *p* = 0.002) of sperm cells was significantly different between the healthy and addicted groups. Protamine-2 gene and protein expression in the addicted group were significantly lower than the healthy group, whilst seminal miRNA-122 levels in addicted men were higher than in healthy men
Opiates/opioids	Nazmara et al.	2019	Iran	Case-control	50	25 heroin abusers 25 healthy controls	Semen quality, sexual hormones levels, sperm histone replacement abnormalities	Sperm motility, viability, and sperm histone replacement abnormalities were significantly different in addicted group vs. non-exposed ones (*p* < 0.05). Serum sex hormone levels were not significantly different between groups
Opiates/opioids	Safarinejad et al.	2012	Iran	Case-control	288	142 heroin abusers 146 healthy controls	Semen quality, sperm function, seminal plasma antioxidant capacity, sperm DNA integrity	The mean ± SD sperm concentration in opiate users and in control subjects was 22.2 ± 4.4 and 66.3 ± 8.3 million per ml, respectively (*p* = 0.002). A significant increase in the amount of fragmented DNA was found in opiate consumers compared with that in controls (36.4 ± 3.8% vs. 27.1 ± 2.4%, *p* = 0.004). Significantly decreased levels of catalase-like and superoxide dismutase-like activity were observed in heroin abusers
Opiates	Bassiony et al.	2020	Egypt	Case-control	90	60 tramadol abusers 30 healthy controls	Semen quality, sexual hormones levels	Tramadol abusers had higher prolactin and lower free testosterone levels and they were more likely to have lower sperm counts and higher levels of abnormal motility and abnormal sperm morphology
Amphetamine-type stimulants (ATS)	Pham et al.	2022	U.S.	Retrospective cohort study	8861	106 men out of the total were prescribed with stimulants for ADHD	Semen quality	Stimulant use was associated with reduced total motile sperm count in the setting of decreased semen volume, but not sperm concentration, motility and morphology
Amphetamine-type stimulants (ATS)	Shalev et al.	2021	Israel	Retrospective cohort study	9769	293 men out of the total were prescribed with methylphenidate	Semen quality	Methylphenidate exposure did not affect sperm morphology but was associated with increased sperm concentration as well as increased total sperm count and total sperm motility among current and past users
Herbal highs/Khat	Hakim et al.	2002	Ethiopia	Prospective cross-sectional	214	184 khat users 30 healthy controls	Semen quality	No statistically significant difference in seminal quality parameters between groups

Abbreviations: CI: confidence interval; DNA: deoxyribonucleic acid; FSH: follicle-stimulating hormone; miRNA-122: microribonucleic acid-122; OR: odds ratio; SD: standard deviation; U.S.: United States; WHO: World Health Organization.

## Data Availability

Not applicable.

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
