# Peer review of "Recreational Drug Misuse and Its Potential Contribution to Male Fertility Levels’ Decline: A Narrative Review"

_brainsci, 2022, doi:10.3390/brainsci12111582_

Round 1

Reviewer 1 Report

The review focused on “Recreational drug misuse and its potential contribution to male fertility levels’ decline: a narrative review”. After careful evaluation, the review compiled the data on steroid drugs that related to spermatogenesis and was written in a good manner. However, need to address the following comments.

Some critical suggestions

1.      Authors should present each molecular mechanism in pictorial form.

2.      The data in the manuscript should be presented in a Table form.

Author Response

The review focused on “Recreational drug misuse and its potential contribution to male fertility levels’ decline: a narrative review”. After careful evaluation, the review compiled the data on steroid drugs that related to spermatogenesis and was written in a good manner. However, need to address the following comments.

Authors: We thank the Reviewer for the overall positive comment to our manuscript.

Q1: Authors should present each molecular mechanism in pictorial form.

A1: Please find attached a range of recreational drugs’ molecular mechanisms original illustrations; thanks for the advice

Q2: The data in the manuscript should be presented in a Table form.

A2: Data have now been presented in table 1; thanks for the advice

Reviewer 2 Report

This is a very interesting paper summarizing evidence of the impact of substance misuse on infertility in men. The paper is well-written and of interest for the journal. However, several minor changes are recommend.

Abstract.

1-The abstract is too long. I recommend to summarize the introduction of the abstract.

2-The methods should be also summarized. This is a non-systematic narrative review. Which databases were used? How were the results presented and analysed? If the results could not be quantitatively described, and were qualitatively presented, this should be described.

Introduction

1- The authors start the paper with the definition of male infertility. However, I consider that the main topic of the paper is the impact of recreational drug misuse. I consider that it has more sense to start the paper with description of the impact of drug use on general health. And in a second step, to focus the introduction on male infertility.

2- The main objective of the current narrative review was to describe the potential effect of drug misuse on male fertility. This paragraph should be rephrased. Where this drugs (e.g. cannabinoids, stimulants, opiates/opioids, and psychedelics), the only drugs to be evaluated?

Material and methods.

1- This section should be divided into several subsections: Screening and selection processes. Inclusion/exclusion criteria, etc. 

Why are the authors only including papers from the last decade? Please, justify it.

Results

1- How many paper were included in the current narrative review? Please describe how many were included and how many for each king of drug misuse. 

2-Why was alcohol not included? If there is a reason, please explain it in the introduction section or "aims subsection".

3-Was male infertility weel-defined in the all the included studies? This should be included in the limitations/strenghts subsection.

Discussion

1- The discussion section is brief. I recommend to include more papers for a deeper discussion of the results. What about other drugs? 

Author Response

This is a very interesting paper summarizing evidence of the impact of substance misuse on infertility in men. The paper is well-written and of interest for the journal. However, several minor changes are recommended.

Authors: We thank the Reviewer for the overall positive comment to our manuscript.

Q1: Abstract. The abstract is too long. I recommend to summarise the introduction of the abstract

A1: thanks for the advice; the abstract has now been reduced from the original number of 407 words to 245; this resulted in a reduction of about 40%.

Q2: Abstract. The methods should be also summarized. This is a non-systematic narrative review. Which databases were used? How were the results presented and analysed? If the results could not be quantitatively described, and were qualitatively presented, this should be described.

A2: Thanks for your advice. To take into account your suggestions, the methodology component of the abstract was here modified as follows: ‘…Whilst searching the MEDLINE/PubMed, a comprehensive overview of the literature regarding male infertility and substances of abuse (e.g. phytocannabinoids, opiates/opioids, stimulants, ‘herbal highs’, psychedelics, and ‘novel psychoactive substances), was here undertaken. Due to the paucity of robust, high quality, empirical, human studies a narrative strategy was here preferred over a systematic approach. Relevant data were qualitatively analyzed and presented in a Table …’

Q3: Introduction. The authors start the paper with the definition of male infertility. However, I consider that the main topic of the paper is the impact of recreational drug misuse. I consider that it has more sense to start the paper with description of the impact of drug use on general health. And in a second step, to focus the introduction on male infertility.

A3: Consistent with your advice, the title of the introductory section 1.1 has been changed as follows: ‘Psychoactive substances’ intake and related medical untoward consequences; Male infertility: definition, causes, and risk factors’. Hence, the following paragraph has been added at the start of this same section: ‘…..The recreational drug intake may be associated with a range of medical untoward consequences. For example, a few cardiotoxic (e.g. dysrhythmias, cardiac arrest, chest pain, and myocardial infarction), effects have been reported for those molecules interacting with the cannabinoid receptor e.g. phytocannabinoids (Ahmad et al, 2022), and especially so for the high-potency synthetic cannabimimetics (Schifano et al, 2017; Zangani et al, 2020). Conversely, the use of ‘uppers’ (including: cocaine; synthetic cathinones; amphetamine-type stimulants) is typically associated with both agitation, ranging from mild agitation to severe psychosis, and a vast range of

sympathomimetic effects (e.g. tachycardia and hypertension as well as psychoactive effects); hyperthermia, rhabdomyolysis, renal failure, and seizures (Schifano et al, 2017). Finally, users of the 5-HT2A agonist psychedelics report effects such as euphoria, mild stimulation, enhanced appreciation of music/light, visually appealing distortions, and altered sense of time and space (Schifano et al, 2017; Catalani et al, 2021).

One of the least mentioned untoward consequences of recreational drug misuse is however its possible association with male infertility….’

Q4: Introduction. The main objective of the current narrative review was to describe the potential effect of drug misuse on male fertility. This paragraph should be rephrased. Where these drugs (e.g. cannabinoids, stimulants, opiates/opioids, and psychedelics) the only drugs to be evaluated?

A4: Consistent with both your advice and in response to the query Q8 as well, the final part of the Introduction has been rephrased as follows: ‘…The current narrative review aims at describing and critically appraise the most recent suggestions of the literature relating to the potential effect on male fertility of most popular recreational molecules, e.g. cannabinoids; stimulants; opiates/opioids; psychedelics; ‘herbal highs’, and novel psychoactive substances (NPS). Regarding alcohol, the molecule may indeed be a component of the recreational drug scene, where it is being ingested in a poly drug consumption pattern. Conversely, alcohol intake is largely prevalent in the general, non-drug misusing, population. Furthermore, a range of high-quality, empirical, studies focussing on the impact of low/moderate/high alcohol intake levels (De Jong et al, 2014; Jensen et al, 2014) have already dealt with this issue. For these reasons, alcohol was not here considered as part of the focus of the current review.

Q5: Material and methods. This section should be divided into several subsections: Screening and selection processes. Inclusion/exclusion criteria, etc. 

A5: Consistent with your advice, the Methodology section has been fully redrafted and better organized, as follows:

Screening and selection process

A comprehensive search was performed on MEDLINE/PubMed, since its inception, to identify those studies which were here considered as both appropriate and representative. The search itself, which was carried out between May and October 2022, focussed on those studies using the terms ‘male infertility’, ‘recreational drug misuse’, ‘cannabinoids’; phytocannabinoids’, ‘opiates’, ‘opioids’, ‘amphetamines’, ‘amphetamine-type stimulants’, ‘stimulants’, ‘cocaine’; ‘methylphenidate’, ‘stimulant plants and herbs’, ‘herbal highs’, ‘LSD’, ‘psychedelics’ ‘novel psychoactive substances’, ‘NPS’, AND ‘male infertility’. A range of keyword strings were used, e.g. drugs AND male infertility, infertility OR among healthy individuals AND recreational drug misuse/abuse/dependence. Resulting abstracts were scrutinised for relevance; the related studies were then summed up after an interactive peer-review process. More in particular, the related papers were here selected for their appropriateness by NS and SC; any disagreement was resolved with the help of a senior researcher (FS).

Inclusion and exclusion criteria

Only articles published in English in peer-reviewed journals were selected. Inclusion criteria related to both quantitative and qualitative studies relating to the use of recreational drugs in humans. Although publications from the past 10 years (e.g. from October 2012 to October 2022) were given the highest levels of priority in the selection process, a number of highly regarded older publications were here included as well. Conversely, studies focusing on children or subjects with medical diagnoses using the selected drugs/substances for medical reasons were excluded from the review.

Data analysis and presentation

With the current strategy, some 98 papers were here identified; however, for 49 of them the main focus referred to molecules (e.g. nicotine; anabolic androgenic steroids; prescribed medications; alcohol) different from those having been identified here as of interest. Conversely, the remaining 49 (e.g. opiates/opioids: 13; cocaine: 1; amphetamine-type substances/remaining stimulants:  4; THC: 27; herbal highs: 2; LSD: 2; NPS: 0) related to the main focus of the review. Interestingly, some 43/49 of these studies had been published over the last decade. However, due to the paucity of robust, high quality, empirical, human studies a narrative strategy was here preferred over a systematic approach. Relevant data were here qualitatively analysed and most representative findings were presented in Table 1.

Q6: Material and methods. Why are the authors only including papers from the last decade? Please, justify it.

A6: Thanks for this meaningful advice, which gave us the opportunity of better explaining the papers’ selection process. In the Methodology section, sub-heading ‘Inclusion and exclusion criteria’, we specified the following: ‘…Although publications from the past 10 years (e.g. from October 2012 to October 2022) were given the highest levels of priority in the selection process, a number of highly regarded older publications were here included as well….’. a few lines below, in the sub-heading ‘Data analysis and interpretation’, we specified the following as well: ‘Interestingly, some 43/49 of these studies had been published over the last decade.…’

Q7: Results. How many papers were included in the current narrative review? Please describe how many were included and how many for each kind of drug misuse. 

A7: Consistent with your advice, the Methodology section will now include the following statement: ‘…With the current strategy, some 98 papers were here identified; however, for 49 of them the main focus referred to molecules (e.g. nicotine; anabolic androgenic steroids; prescribed medications; alcohol) different from those having been identified here as of interest. Conversely, the remaining 49 (e.g. opiates/opioids: 13; cocaine: 1; amphetamine-type substances/remaining stimulants: 4; THC: 27; herbal highs: 2; LSD: 2; NPS: 0) related to the main focus of the review.….’

Q8: Results. Why was alcohol not included? If there is a reason, please explain it in the introduction section or "aims subsection".

A8: Thanks for your meaningful advice. Consistent with this, at the end of the Introduction section, the following statement was added: ‘…..Regarding alcohol, the molecule may indeed be a component of the recreational drug scene, where it is being ingested in a poly drug consumption pattern. Conversely, alcohol intake is largely prevalent in the general, non-drug misusing, population. Furthermore, a range of high-quality, empirical, studies focussing on the impact of low/moderate/high alcohol intake levels (De Jong et al, 2014; Jensen et al, 2014) have already dealt with this issue. For these reasons, alcohol was not here considered as part of the focus of the current review….’

Q9: Results. Was male infertility well-defined in all the included studies? This should be included in the limitations/strenghts’ subsection.

A9: Consistent with your advice, the whole Discussion has been restructured. Within the Discussion a specific section (‘Male infertility measurement; methodological issues‘) has now been included; the section now starts with the following statements: ‘…Most empirical studies mainly focussed here on semen analysis (e.g. sperm concentration, motility, viability, and morphology), but this biological parameter, per se, is not an ideal discriminator of male fertility. Furthermore, as very recently suggested, there is a lack of knowledge about possible molecules used as candidates for the early identification of male infertility risk (Omes et al, 2022). It is of interest to note that the different studies here commented did refer to a range of different parameters to define male infertility (e.g. abnormal morphology; abnormal motility; low sperm volume; presence of asthenozoospermia and/or teratozoospermia; white blood cell count in semen and survival rate of sperm cells etc; see Table 1) and this may have introduced here some biases and related difficulties in the overall interpretation of findings……’.

Q10: The discussion section is brief. I recommend to include more papers for a deeper discussion of the results. What about other drugs?

A10: Consistent with your advice, the whole Discussion has been restructured. A further 9 references, between the Introduction and the Discussion, have now been added.

Some statements have now been included as well, such as: ‘It is however necessary to keep in mind that drug abusers are different from non-users from a range of perspectives, including higher odds of unhealthy lifestyle habits such as alcohol, tobacco use, and caffeine[22]. For example, the relationship between caffeine consumption and infertility in the general population is unclear; hence, Bu et al (2020) carried out a systematic review of the evidence from any type of controlled clinical studies to explore whether caffeine intake is a risk factor for human infertility. Their results showed that low (OR 0.95, 95% CI 0.78-1.16), medium (OR 1.14, 95% CI 0.69-1.86) and high doses (OR 1.86, 95% CI 0.28-12.22) of caffeine intake may not increase the risk of infertility. They suggested as well, however, that the quality of the current evidence bodies were all low. In contrast with this, a systematic review assessing the role of caffeine intake on male infertility concluded that caffeine intake, possibly through sperm DNA damage, may negatively affect male reproductive function (Ricci et al, 2017). The Authors suggested as well, however, that the evidence from epidemiological studies on semen parameters and fertility is inconsistent and inconclusive……’. Overall, the length of the Discussion stands now at about 1,000 words.

Round 2

Reviewer 1 Report

The manuscript was revised carefully according to the reviewer's comments. Moreover, I appreciate the author's work.